# Hyaluronic Acid Alleviates Oxidative Stress and Apoptosis in Human Tenocytes via Caspase 3 and 7

**DOI:** 10.3390/ijms23158817

**Published:** 2022-08-08

**Authors:** Marialucia Gallorini, Cristina Antonetti Lamorgese Passeri, Amelia Cataldi, Anna Concetta Berardi, Leonardo Osti

**Affiliations:** 1Department of Pharmacy, University “G. d’Annunzio” Chieti-Pescara, Via dei Vestini 31, 66100 Chieti, Italy; 2Laboratory of Stem Cells, Department of Haematology, Transfusion Medicine and Biotechnologies, Santo Spirito Hospital, 65124 Pescara, Italy; 3Unit of Arthroscopy and Sports Medicine, Hesperia Hospital, 41125 Modena, Italy

**Keywords:** hyaluronic acid, molecular weight, tendon, caspase, oxidative stress, inflammation, apoptosis, ascorbic acid, collagen, nitic oxide

## Abstract

Rotator cuff tendinopathy (RCT) is the primary reason for shoulder surgery and its clinical management is still challenging. Hyaluronic acid (HA) has been shown to have anti-inflammatory effects in vitro and in vivo under RCT conditions, characterized by an exaggerated oxidative stress (OS). However, molecular mechanisms underlying HA-related effects are still partially disclosed. With these aims, a cell model of RCT was established by exposing primary human tenocytes to H_2_O_2_ for up to 72 h. Four different HAs by molecular weight were administered to measure nitric oxide (NO) and OS, apoptosis, and collagen 1 expression. In parallel, the well-known antioxidant ascorbic acid was administered for comparison. The present study highlights that HAs characterized by a low molecular weight are able to counteract the H_2_O_2_-induced OS by decreasing the percentage of apoptotic cells and reversing the activation of caspase 3 and 7. Likewise, NO intracellular levels are comparable to the ones of controls. In parallel, collagen 1 expression was ameliorated by HAs characterized by higher molecular weights compared to AA. These findings confirm that HA plays an antioxidant role comparable to AA depending on the molecular weight, and highlight the molecular mechanisms underlying the HA anti-apoptotic effects.

## 1. Introduction

Rotator cuff tendinopathy (RCT) is the most common cause of shoulder pain, with an increasing frequency which varies from 5% to 40% [1,2]. RCT is characterized by tendon structure disruption, ineffective neovascularization, and decreased collagen 1 production [3]. A role in the pathogenesis of tendinopathies for oxidative stress (OS) has been described in several studies [4]. It has been widely demonstrated that the accumulation of reactive oxygen species (ROS) stimulates the intracellular antioxidant system, which is tightly regulated by Nrf2 (nuclear factor erythroid 2 (NF-E2)-related factor 2) [5], and that a functional tendon-repair failure might be strongly associated with the activation of the Nrf2 pathway [6]. Recently, hyaluronic acid (HA) has been suggested for the clinical management of tendinopathies [7,8]. Our previous studies demonstrated that different HA preparations, depending on their molecular weight, counteracted apoptosis under H_2_O_2_-induced OS [9], enhanced cell recovery from H_2_O_2_ exposure in terms of decreased cytotoxicity and reduced Nrf2 expression, and increased the expression of the HA receptor CD44 [10].

HA, as one of the fundamental components of cartilage and tendons, contributes to their viscoelastic properties in vivo [8,11,12]. However, HA efficacy in the management of RCT is still debated and a comprehensive description of the underlying molecular mechanisms involved is still lacking [8].

In this light, to better understand the different HA biological outcomes under OS conditions, and to extend and improve HA pharmaceutical applications, the antioxidant effects of four different HAs by molecular weight on human rotator-cuff tendon-derived cells in terms of cytoprotection and caspase 3/7 activation were studied. Moreover, protein expression levels of collagen 1, the predominant collagen isoform involved in ECM deposition under inflammatory conditions, was investigated during HA administration. In addition, to evaluate whether the HA might have antioxidant functions, the well-known antioxidant ascorbic acid (AA) was used for comparison [13]. Moreover, AA is well known to participate in collagen biosynthesis [14], including in an Achilles tendon injury rat model [15].

## 2. Results

### 2.1. Effect of HAs on Cell Proliferation Rate and Viability

A cell proliferation analysis for up to 72 h of treatment was performed on tenocytes to verify the effects of HAs administered in parallel with H_2_O_2_ on their proliferation rate (Figure 1A). As expected, the percentage of H_2_O_2_-exposed proliferating tenocytes dramatically fell after 24 h, and the oxidative stressor administered had an even greater effect after 72 h, with proliferation percentages less than 5% in all the samples. When AA and HA preparations were administered in parallel with H_2_O_2_ after 24 h, the proliferation rate slightly but significantly increased in all the experimental conditions with respect to H_2_O_2_ alone. After 72 h, this effect was maintained.

### 2.2. Modulation of Superoxide Anions in the Presence of HAs

The intracellular generation of superoxide anions was measured in tenocytes after 6 h of exposure in all the experimental conditions (Figure 1B). When H_2_O_2_ is added to the cultures, a dramatic increase in the superoxide anion generation is registered, with the MFI ratio being 3-folds more than the one of the UC (Figure 1B). As expected, AA is able to significantly reduce the generation of superoxide in the presence of H_2_O_2_, with the MFI ratio halved with respect to H_2_O_2_ alone. In parallel, Hyalgan is the best in decreasing the amount of superoxide anions among the four HA preparations, followed by Hyalubrix and Artrosulfur HA.

### 2.3. Oxidative Stress-Induced Apoptosis Counteraction by HAs

Cell percentages in the various phases of cell death were assessed after 48 h in the presence of HAs under OS conditions (Figure 1C). After 48 h, viable cells appeared time-dependently decreased by H_2_O_2_ (38.37%) and early apoptotic cells dramatically increased (39.14%), as did late apoptotic tenocytes. When AA is added, cell viability significantly increases (56.60%) for late apoptotic cells and necrotic cells, whereas the percentage of early apoptosis is comparable to the one of H_2_O_2_ alone. In parallel, Artrosulfur HA and Synolis VA increase cell viability, decrease early apoptosis, but increase cell necrosis with respect to H_2_O_2_ alone (11.51%), with percentages measured as 21.31% and 24.57%, respectively. Cell viability is even more increased in the presence of Hyalubrix and Hyalgan (65.49% and 68.39%, respectively). In parallel, apoptosis is slightly decreased, and viability is slightly increased.

### 2.4. HAs Treatment Reduces the Activation of Caspase 3 and 7

To establish whether the apoptosis induction might be driven by the activation of caspases, the activity of caspase 3 and 7 was measured after 48 h under OS conditions (Figure 1D). By using the image-analyzing software, the mean fluorescent intensity related to caspase-positive (apoptotic) cells was measured (see Section 4). Compared to the UC control (MFI = 1.07), the H_2_O_2_ exposure dramatically increases the activation of caspase 3 and 7 (MFI = 14.46). In parallel, when the AA is added to the culture, the caspase 3 and 7 MFI decreases (5.69), indicating a reduced activity of caspases. Artrosulfur HA and Synolis VA slightly reduce caspase activity (MFI = 4.76 and 4.46, respectively) compared to AA, as shown in Figure 1D. When Hyalubrix and Hyalgan are administered, the activity of caspase 3 and 7 was even more reduced (MFI = 2.46 and 2.23, respectively) compared to AA. This suggests that HA counteracts the H_2_O_2_-induced apoptosis through the reduction of caspase 3 and 7 activation.

### 2.5. Modulation of Nitric Oxide (NO) in the Presence of HAs

The intracellular nitric oxide generated by tenocytes was measured in tenocytes after 6 h of exposure in all the experimental conditions (Figure 2A). H_2_O_2_ exposure triggers the NO fall (Figure 2A). When AA and the HA preparations are added, NO is slightly decreased with respect to UC, but it is maintained higher than H_2_O_2_ alone, mainly with Hyalubrix and Hyalgan.

### 2.6. Modulation of Collagen 1 Levels by HA Preparations

Collagen 1 expression levels were measured in tenocytes under OS conditions after 72 h (Figure 2B). As expected, the H_2_O_2_ exposure strongly decreases collagen 1 expression levels compared to the UC, whereas the presence of AA ameliorates collagen 1 expression levels under OS conditions. Data reveal that Hyalgan exerts a greater effect on tendon-derived cells exposed to H_2_O_2_ than all the other treatments in terms of collagen 1 expression.

## 3. Discussion

Oxidative stress (OS) is crucial under tendon injuries [16]. This might be attributed to the links among OS, inflammation, and apoptosis, all of which impair the structure and function of cells. An excessive OS results in the activation of redox-sensitive signal transduction pathways [17].

In the present work, OS was induced on tendon-derived cells by the H_2_O_2_ exposure. Our group already demonstrated that the HA could counteract apoptosis under OS induced by H_2_O_2_ [9]. Here, our previous results are consolidated, showing that all the four HAs used significantly counteracted the cytotoxic action of H_2_O_2_ in terms of proliferation (Figure 1A).

Since intracellular ROS is mostly produced in mitochondria, the superoxide mitochondrial production was measured to verify whether the HA might influence its modulation. Here, although all the HA preparations decrease the amount of superoxide anions, Hyalgan is revealed as the best preparation in counteracting their production (Figure 1B). Thus, our results suggest that HAs characterized by a lower molecular weight reduce the production of mitochondrial ROS.

Caspase 3 is a key protein in the stimulation of apoptosis, being the primary executioner in apoptotic cell death. In parallel, caspase 7 plays a supportive role in the execution phase of the process [18]. Therefore, we applicated a caspase 3/7 visualization to evaluate the HA efficacy in terms of anti-apoptotic cell responses under OS conditions [19]. Very interestingly, all the HAs have a similar behavior (Figure 1D). Consequently, it was highlighted that HA ameliorates H_2_O_2_-induced apoptosis via caspase 3/7, independently from the molecular weight.

The generation of NO plays a relevant role in repairing an injured tendon [20,21]. In addition, the use of a NO-releasing pro-drug was proven to be useful in the development of promising NO-based therapy [22]. For these reasons, the evaluation of NO expression is an important parameter to be considered in damaged tendons. Firstly, the H_2_O_2_ exposure caused a decrease in the production of NO. It is here shown that HAs induced a significant increase in the NO production compared to H_2_O_2_ alone-treated cells (Figure 2A). In detail, Artrosulfur, Hyalubrix, and Hyalgan were able to induce a higher NO production than the very well-known antioxidant AA [23]. This result is particularly intriguing since it was demonstrated that NO favors tendon healing at the molecular level by increasing collagen synthesis after 7 days of treatment in an in vitro tendon cell culture [24]. However, to offer deeper insights into the NO effects in tendinopathies, further studies are necessary.

Collagen is the major molecule secreted by tendon cells, participating in ECM remodeling and maintenance [25]. Physiologically, tendons are composed of 65–85% of collagen 1 [14]. In our previous studies, we already demonstrated that Synolis VA was able to stimulate collagen 1 synthesis in a dose dependent manner over 14 days [9]. In the present work we investigated the effect of the HA treatment at an earlier exposure time (72 h). Our results show that collagen 1 expression is downregulated during the H_2_O_2_ treatment. Under OS conditions, Hyalubrix and Hyalgan ameliorate the expression of collagen 1. It can be therefore speculated that these two HAs act earlier with respect to Synolis VA in terms of collagen 1 deposition, and this effect might be related to the difference in the molecular weight, as Synolis VA is the highest one.

In conclusion, the OS-induced apoptosis in tenocytes is driven by caspase 3 and 7, and this condition could be counteracted by the administration of HAs. Moreover, the administration of HAs with lower molecular weights leads to an earlier collagen 1 deposition and displays better outcomes in terms of OS counteraction, comparable to the well-known antioxidant AA. This in vitro study can lay the grounds for better outcomes regarding the clinical management of RCTs.

## 4. Materials and Methods

### 4.1. Cell Culture

Cryopreserved, tendon-derived cells, stored in vials in liquid nitrogen and isolated from the same patients used in our previous work, were used [26]. The cell isolation protocol was extensively described previously [9,10,26,27]. The tenocyte phenotype was confirmed by assessing the expression of tenocyte-specific genes [28]. For the present work, cells at passage 0 from three samples were thawed out and promptly cultured to avoid phenotype changes caused by later passages. Tenocytes were cultured as elsewhere [26] and used up to passage 5.

### 4.2. Cell Treatment

Tenocytes were seeded in 96- (0.5 × 10^4^ cells/well), 12- (0.2 × 10^5^ cells/well) or 6-well (0.5 × 10^5^ cells/well) tissue culture-treated plates (Falcon^®^, Corning Incorporated, Brooklyn, NY, USA) and were left to adhere for 24 h. Cultured cells were exposed to 2 mM of H_2_O_2_ (stock solution 30% *v*/*v*, Sigma-Aldrich, Milan, Italy), ascorbic acid (AA), 50 μg/mL (stock powder purchased from Sigma-Aldrich, Milan, Italy), and to four HAs, namely, Artrosulfur HA^®^ (M.W. 1000 KDa) (HA1), Synolis VA^®^ (M.W. 2200 KDa) (HA2), Hyalubrix^®^ (M.W. 1600 KDa) (HA3), and Hyalgan^®^ (M.W. 500–730 KDa) (HA4) at a final concentration of 1 mg/mL as reported previously [9,10]. Untreated cells (UC) were used as the control. Exposure times varied according to the various parameters analyzed, from 6 h up to 72 h.

### 4.3. Proliferation Assay (alamarBlue)

At the established exposure times (24 or 72 h), the medium was replaced with a fresh one containing 10% alamarBlue reagent (Invitrogen, Thermo Fisher Scientific, Waltham, MA, USA), and afterwards incubated for 4 h at 37 °C. The absorbance was measured at 570 and 600 nm by means of a Multiskan GO microplate spectrophotometer (Thermo Fisher Scientific). The value obtained without cells was established as the negative control.

### 4.4. Detection of Mitochondrial Superoxide Anions

After 6 h of exposure, the generation of mitochondrial superoxide anions was determined by flow cytometry using the fluorescent probe MitoSOX™ (MitoSOX™ Red Mitochondrial Superoxide Indicator, Invitrogen, Thermo Fisher Scientific, Waltham, MA, USA) based on an established procedure as elsewhere reported [29]. Relative fluorescence emissions were detected by a CytoFLEX flow cytometer (Beckman Coulter, Indianapolis, IN, USA) and they were expressed as mean fluorescence intensity ratios on the unstained control (not shown).

### 4.5. Determination of Apoptosis

The PE Annexin V/Dead Cell Apoptosis Kit with SYTOX^®^ Green for Flow Cytometry (Invitrogen, Thermo Fisher Scientific, Waltham, MA, USA) was used to detect apoptosis by flow cytometry after 24 or 48 h of exposure as reported elsewhere [9]. Cell populations were detected by a CytoFLEX flow cytometer (Beckman Coulter, Indianapolis, IN, USA) and analyzed with the CytExpert software (Beckman Coulter, Indianapolis, IN, USA).

### 4.6. Detection of Caspase 3 and 7

Active caspase 3 and 7 were detected after 48 h of exposure using the Image-iT™ LIVE Red Caspase 3 and 7 Detection Kit (Invitrogen, Molecular Probes, Waltham, MA, USA), following the manufacturer’s instructions. The kit employs a novel approach to detect active caspases that is based on a fluorescent inhibitor of caspase (FLICA™) methodology [30]. Samples were observed through an ECLIPSE Ti-U inverted fluorescent microscope (Nikon Instruments Inc., Melville, NY, USA) equipped with a camera. For the image analysis, all digital images were captured with the NIS-Elements Imaging Software (Nikon Instruments Inc.). The total fluorescence intensity of an area ≥ 10 frames from each slide was determined. The intensity level was normalized with one of the untreated controls.

### 4.7. Detection of Intracellular Nitric Oxide

The Nitrixyte™ probe was used for detecting free NO in tenocytes (Cell Meter™ Fluorimetric Intracellular Nitric Oxide Assay Kit; AAT Bioquest, Inc., Sunnyvale, CA, USA) by flow cytometry as already reported [21]. Relative fluorescence emissions were detected by a CytoFLEX flow cytometer (Beckman Coulter, Indianapolis, IN, USA) and analyzed with the CytExpert software (Beckman Coulter, Indianapolis, IN, USA), and they were expressed as mean fluorescence intensity ratios on the unstained control (not shown).

### 4.8. Expression of Collagen 1 by Flow Cytometry

At the established exposure time (72 h), cell pellets were fixed, permeabilized, and stained as already reported [31]. Next, the mouse monoclonal anti-collagen 1A1 (Thermo Fisher Scientific, Waltham, MA, USA) antibody was added 1:50 to each sample and incubated for 1 h at 4 °C. Next, the secondary phycoerythrin (PE) horse anti-mouse IgG (H + L) antibody (Vector Laboratories, Inc., Burlingame, CA, USA) was added (1:50) and incubated at 4 °C in the dark for 45 min. Relative fluorescence emissions were detected as previously described [31].

### 4.9. Statistics

Statistical analysis was performed using the GraphPad 5.0 software (GraphPad Software, San Diego, CA, USA) by means of a t-test and ordinary one-way ANOVA, followed by post hoc Tukey’s multiple comparisons tests. Values of *p* < 0.05 were considered statistically significant.

## 5. Conclusions

Sustained OS leads to a decrease in tenocyte proliferation caused by an increase in cell death through apoptosis and a consequent diminished ECM remodeling due to a lack of collagen 1 deposition. In this light, understanding that the molecular pathway underlying the OS-induced apoptosis in tenocytes is driven by caspase 3 and 7, and that this condition could be counteracted by HAs characterized by lower molecular weights, can lay the groundwork for better outcomes about the management of RCTs.

## Figures and Tables

**Figure 1 ijms-23-08817-f001:**
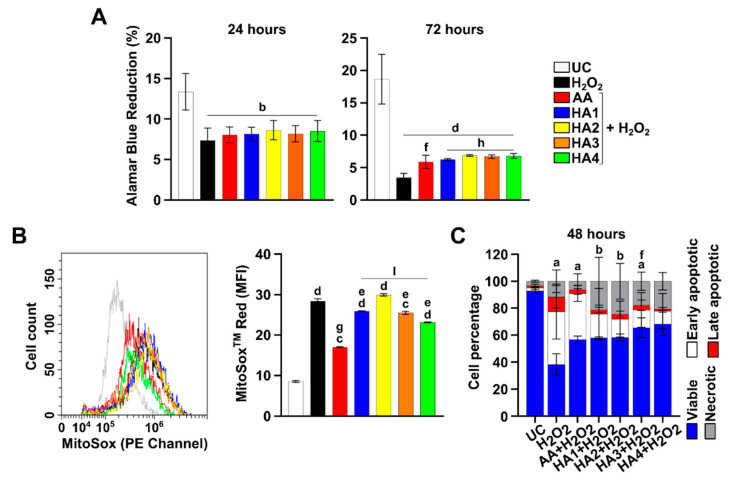
Cell proliferation, generation of superoxide anions, apoptosis, and caspase 3/7 activity in tenocytes exposed to hyaluronic acids under OS conditions. (**A**) Bar graphs represent the % of alamarBlue reduction values after 24 and 72 h. Data are the means (±SD) of three different experiments. (**B**) Overlays of the peaks in the FL2 channel represent the fluorescent emissions of each sample. The bar graph shows the mean fluorescence intensity (MFI) related to the emission in the FL 2 channel which is proportional to the generation of superoxide anions. Values are the ratios of the MFIs generated from each sample on the unstained control (negative). (**C**) Bars represent the percentage of viable cells (unstained), cells in early apoptosis (annexin), cells in late apoptosis (annexin + SYTOX Green), and cells in necrosis (SYTOX Green). (**D**) Representative images from three independent experiments of caspase 3/7 activity (red cells). 1 cm = 100 μm. The bar graph represents the MFI related to stained cells for activated caspases. Untreated control (UC); hydrogen peroxide 2 mM (H_2_O_2_); ascorbic acid μg/mL (AA); Artrosulfur HA^®^ (HA1), Synolis VA^®^ (HA2), Hyalubrix^®^ (HA3) and Hyalgan^®^ (HA4) at a final concentration of 1 mg/mL. a: *p* ≤ 0.05; b: *p* = 0.001; c: *p* = 0.0001; d: *p* < 0.0001 versus UC; e: *p* ≤ 0.05; f: *p* = 0.001; g: *p* = 0.0001; h: *p* < 0.0001 versus H_2_O_2_; l: *p* < 0.0001 versus AA.

**Figure 2 ijms-23-08817-f002:**
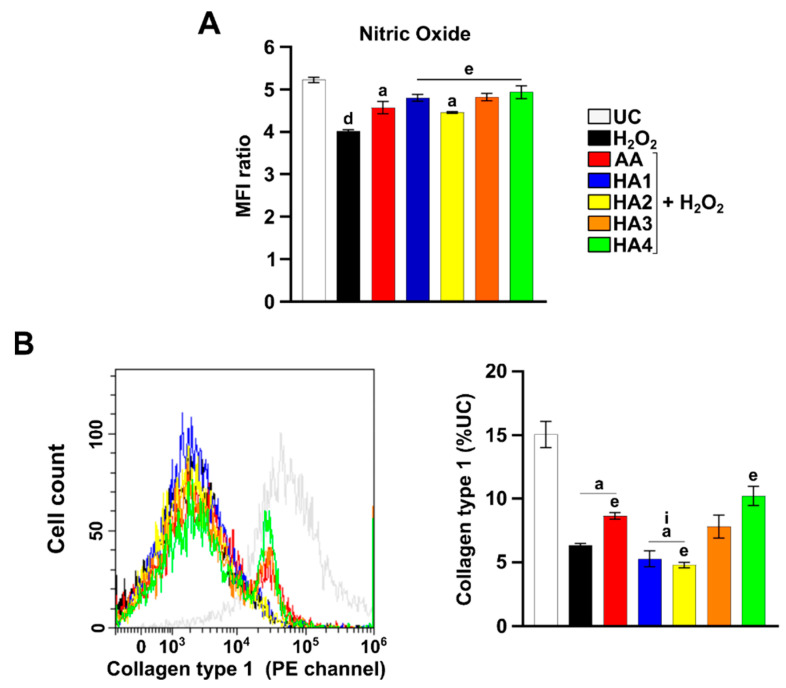
Generation of nitric oxide and collagen 1 expression in tenocytes exposed to hyaluronic acids under OS conditions. (**A**) The bar graph shows the mean fluorescence intensity (MFI) related to the emission in the FL2 channel, which is proportional to the generation of nitric oxide. Values are the ratios of the MFI generated from each sample on the unstained control (negative). (**B**) Overlays of the peaks in the FL2 channel represent the fluorescent emissions of each sample. The bar graph represents the % of expression related to collagen 1. Data are expressed as the % of the UC. Untreated control (UC); hydrogen peroxide 2 mM (H_2_O_2_); ascorbic acid μg/mL (AA); Artrosulfur HA^®^ (HA1), Synolis VA^®^ (HA2), Hyalubrix^®^ (HA3), and Hyalgan^®^ (HA4) at a final concentration of 1 mg/mL. a: *p* ≤ 0.05; d: *p* < 0.0001 versus UC; e: *p* ≤ 0.05 versus H_2_O_2_; i: *p* ≤ 0.05 versus AA.

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
