# Peer review of "Hyaluronic Acid Alleviates Oxidative Stress and Apoptosis in Human Tenocytes via Caspase 3 and 7"

_ijms, 2022, doi:10.3390/ijms23158817_

Round 1

Reviewer 1 Report

The authors aimed to characterize the antioxidant and anti-apoptotic effects of HA (MW between 500 and 2200) on H2O2-treated tendinocytes. The authors reported that:

1) HA has antioxidant and anti-apoptotic activities. HA may also stimulate a synthesis of type 1 collagen. These activities may be influenced by its molecular weights.

2) This study is an in vitro study, and current data are insufficient to support a role of HA in ameliorating rotator cuff tendinopathy. In vivo studies are warranted.

The following questions should be considered:

-      1) Does HA alone stimulate tendinocyte proliferation?

2) What are the expression levels of CD44? Does it play a role in the synthesis of type 1 collagen?

3) What are the cellular levels of peroxynitrite in relation to superoxide and nitric oxide levels?

4) Line 107-108, “In parallel, apoptosis and necrosis are slightly decreased.” Could the authors point to the evidence that leads to the statement?

Reviewer 2 Report

In the report: “Hyaluronic acid alleviates oxidative stress and apoptosis in human tenocytes via caspase 3 and 7” the authors discussed about the biological effects of HA to counteract the oxidative stress into an in vitro model of tenocytes.

Overall, this manuscript results very interesting, the authors clearly explain the rational of the study and discussed the topic point by point.

However, we would like to invite the authors  to clarify some minor points:

1.       Please check the check punctuation and spaces;

2.       Among the introduction section, the authors describes in general the anti-oxidative of HA and its role into maintenance of cartilage mechanical properties. The authors should deep the description of HA application in the field of cartilage and related pathologies.

3. The authors performed the experiments on cryopreserved human tenocytes, please insert some details about the preservation of their penothype. Specific biomarkers was checked before the use?

4.       The authors checked the expression of Caspases also by immunofluorescence? Please, give more details about the used kit in the materials and methods.  It was not possible perform other assay such as western blotting?

5.       The resolution of Figure 1D is not so good, the color appears confused with background in the some pictures. There are other pictures available? Could be improved without create articfacts?

Round 2

Reviewer 1 Report

No further comment.